# Can corporate ESG performance improve audit efficiency?: Empirical evidence based on audit latency perspective

**Li Zhang**[1]*, **Caixia Guo**[2]

**1** School of Accountancy, Tianjin University of Commerce, Tianjin, China, **2** Tianjin University of Commerce, Tianjin, China

* andreyignke@mail.com

**Data Availability Statement:** All relevant data are within the manuscript and its Supporting Information files.

**Funding:** This study was funded by the Tianjin 2020 Philosophy and Social Science Planning Project, "Research on Auditing Financial Funds and

## Abstract

Environmental, Social and Governance (ESG) is closely related to the "dual carbon" objective and the concept of sustainable development. The impact of ESG performance on audit efficiency, especially on audit delays, is still an issue to be studied in depth. Drawing on stakeholder theory, sustainable development theory, shared value concept and corporate social responsibility theory, this study adopts regression analysis and structural equation modeling (SEM) to investigate the impact of ESG on audit efficiency based on the data of A-share listed companies in the period of 2015–2022, with a focus on audit delay. The results of regression analysis show that ESG performance has a significant effect on reducing audit delay, and audit delay is reduced by 0.007 on average for each unit increase in ESG performance. In structural equation modeling, the effect of ESG performance on audit delay is more significant, with an estimated value of -0.555 and a standard error of 0.097. In addition, the study shows that the corporate ESG performance on audit efficiency has a positive impact is more pronounced among firms with stronger ESG practices, especially among non-state-owned firms with lower institutional investor ownership and firms audited by "Big Four" firms. These results not only demonstrate the importance of ESG performance in improving audit efficiency, but also provide important guidance for corporate management and policy making. This study enriches the existing literature on corporate ESG performance and audit efficiency and provides new perspectives and directions for future research.

## 1. Introduction

In the report of the Twentieth National Congress, it was proposed that "green development should be promoted, and harmonious coexistence between human beings and nature should be fostered". Enterprises, as critical components of market economic activities, are increasingly scrutinized by the public within the context of achieving the "dual-carbon" goals and sustainable development. There is growing concern regarding whether these enterprises are fulfilling their social responsibilities and integrating the concept of sustainable development into their

Social Donations for Public Emergencies" (Project No. TJGL20-017). The funders had no role in study design, data collection and analysis, decision to publish, or preparation of the manuscript.

**Competing interests:** All authors hereby declare that there are no financial or other personal conflicts of interest related to this study.

pursuit of economic gains. In 2004, the United Nations Global Compact first introduced the concept of ESG, focusing on three pillars: environmental (E), social (S), and corporate governance (G). Although China's ESG development initially lagged behind international standards, it has seen significant growth in recent years, bolstered by pertinent policies and robust government advocacy. In 2010, the Ministry of Environmental Protection (MEP) issued the Guidelines for Environmental Information Disclosure by Listed Companies, mandating that heavily polluting industries include environmental reports in their annual disclosures, whereas other industries were encouraged to do so voluntarily [1]. In 2015, the Central Committee of the Communist Party of China and the State Council released the Overall Plan for the Reform of the Ecological Civilization System, which called for the progressive creation of a compulsory environmental information disclosure system for listed companies. The revised 'Guidelines on the Content and Format of Information Disclosure for Companies Issuing Securities (Public Issue No. 2—Content and Format of Annual Report)' were released in June 2021, incorporating additional requirements for disclosing a company's environmental, social responsibility, and corporate governance practices [2]. Data from the China Association of Listed Companies indicates that over 1,700 listed companies, representing 34% of the total, have independently prepared and published ESG-related reports for 2022. This represents a significant increase from the 1,112 enterprises disclosing ESG information in the previous year, with a net addition of nearly 600 companies [3]. Despite the lack of standardized ESG rating standards in China, addressing ESG concerns remains a "must-answer" issue for enterprises going forward, and their ESG performance continues to be a focal point within the capital market.

Audit delay represents the time interval between the balance sheet date and the audit report date, reflecting the auditor's audit duration and efficiency. The audit report constitutes the result of a comprehensive assessment by the auditor and serves as a vital channel for investors, creditors, and other stakeholders to gain insights into the listed company. When the audit report is issued after an extended period, that is, when there is significant audit delay, it impairs the timeliness of the information conveyed, detracting from its utility for stakeholders. Moreover, audit delay is a variable associated with audit efficiency, and, when controlling for external factors beyond the auditor's control, it can serve as an indicator of audit efficiency [4, 5]. Existing research suggests that ESG ratings capture the attention of auditors and are incorporated into audit report decision-making [6]. The higher a firm's ESG performance, the greater its reflection in the audit report decisions. Furthermore, a superior ESG performance by a company increases the likelihood of receiving a standard audit opinion and is associated with a significant reduction in audit fees. Currently, there is a scarcity of literature that directly examines the relationship between corporate ESG performance and audit efficiency, and although studies have been conducted to analyze the relationship between ESG performance and audit efficiency, there is still a significant gap in research from the perspective of audit delay.

In view of this, this study aims to fill this gap by adopting regression analysis as well as structural equation modeling in its methodology to explore the relationship between ESG performance and audit delay in depth, while this study also employs the one-period lagged method, Heckman's two-phase method, the double cluster adjustment method, and the method of controlling for other variables to conduct a robustness test. In terms of sample and variable selection, this study selects A-share listed companies from 2015–2022 as the research sample to empirically examine the impact of corporate ESG performance on audit efficiency from the perspective of audit delay, while considering factors such as institutional investor shareholding ratio, audit quality, and the nature of property rights, which provides a more comprehensive perspective for understanding the impact of ESG performance on audit efficiency. Compared with the existing literature, the contribution of this study is mainly in the

following three aspects: first, this study broadens the scope of research in the field of auditing related to ESG performance. Second, focusing on audit delay, it explores the impact of corporate environmental, social and governance performance on audit efficiency. Third, at the practical level, it provides valuable insights for auditors to perform audit tasks.

The overall structure of the paper is organized as follows:

The first part is the introductory part, which introduces the relevant background and importance of the study; the second part reviews the relevant literature as well as the theoretical analysis and research hypotheses, which provides important theoretical support for the main body of the thesis based on pointing out the shortcomings and research gaps of the existing studies; the third part describes the design of the study, indicating the source of the samples, the definition of the relevant variables, as well as the research methodology adopted; the fourth part describes the research The fourth part describes the results of the study, using data to prove the research hypotheses; the fifth part is the discussion, focusing on the theoretical and practical significance of the findings; the sixth part is the conclusion and policy recommendations, based on the findings of the existing research, this study puts forward practical policy recommendations.

## 2. Literature review

### 2.1 ESG performance

In the context of the "dual-carbon" goal and the emerging development concept, companies are not the only entities attentive to their own "sustainable development"; stakeholders are increasingly focusing on the ESG performance of investee companies as well. The existing literature primarily concentrates on the financial accounting domain, with studies demonstrating that corporate ESG performance can incentivize enterprises to boost R&D investment, augment innovation output, enhance the number of green innovations, and bolster sustainability performance in the manufacturing industry [7–9]. Moreover, firms with superior ESG performance tend to incur lower debt financing costs, mitigate financial risks, positively influence surplus sustainability, and realize future stock returns that are significantly higher than those of their lower-rated counterparts [10–12]. Furthermore, the disclosure of ESG information appears to be more critical for financial performance than the actual implementation of ESG practices, and the extent of variation in assessments by different rating agencies can influence a firm's stock price [13–15]. Thus, it is evident that a firm's ESG performance can impact various aspects, including financial risk.

In auditing research, corporate ESG performance can offer stakeholders insights into a company's operations, aid auditors in understanding the audited entity, and help identify risks of material misstatement. Existing studies examine the economic consequences of ESG performance from various angles. Regarding audit fees, corporate ESG performance is found to mitigate information and operational risks, while good ESG performance lowers both by easing financing constraints and enhancing corporate transparency [16, 17]. Concerning audit opinions, ESG ratings of listed companies may affect the issuance of audit reports as firms with strong ESG performance present lower risks of material misstatement and operational issues, leading auditors to be more likely to issue unqualified audit opinions [18, 19].

Additionally, research has shown that media attention, investor focus, and party organization governance all play a part in enhancing firm ESG performance via the oversight function of external stakeholders. Co-institutional investors may further boost corporate ESG performance by leveraging governance mechanisms and collaborative effects. Furthermore, Geiger & Kumas (2023) suggest that the growth of the urban digital economy positively correlates with enhanced corporate ESG performance [20].

## 2.2 Audit delays

Audit delay is an essential indicator of audit efficiency, reflecting the auditor's input. A firm's reputation is inversely associated with audit report lag, earnings announcement delay, and the probability of announcing earnings post-audit completion [21]. Furthermore, audit delays tend to be extended when the CEO is promoted internally. When public companies acquire directors' and officers' liability insurance, subsequent reductions in audit delays and improvements in audit efficiency are observed [22].

At the accounting firm level, industry growth has led to increased informatization within firms, benefitting both audit quality and efficiency. The consolidation of accounting firms impacts audit efficiency, with a noticeable increase in audit delays for firms one year post-consolidation compared to their non-consolidated counterparts. Knowledge sharing within firms is correlated with enhanced audit efficiency, evidenced by shorter audit lags For individual auditors, audit latency diminishes as the signing accountant accrues more practice experience [23]. Auditors specializing in specific industries can utilize their expertise to rapidly acquaint themselves with a client's business operations, often completing audits more swiftly than non-specialists. Additionally, the high-speed rail (HSR) pass-through and delisting system may contribute to reduced audit delays, whereas negative media coverage is linked to audit pricing without a significant correlation to audit delay [24].

In summary, while scholars have conducted in-depth discussions on ESG performance and audit delay independently, a direct analysis of the relationship between the two has not yet been undertaken. Theoretically, companies exhibiting strong ESG performance signal higher governance standards and superior management, potentially reducing audit risk and enhancing audit efficiency. Accordingly, this study aims to further investigate the impact of corporate ESG performance on audit efficiency, thereby enriching the body of existing literature.

## 3. Theoretical analysis and research hypotheses

Stakeholders are increasingly concerned about corporate ESG performance, as it can enhance stakeholder satisfaction and reduce corporate risk according to Stakeholder Theory. Positive ESG performance can lead to trust and recognition from stakeholders, fostering long-term and stable relationships. Improving ESG performance helps companies strengthen their information disclosure systems and risk management measures. Through external supervision, stakeholders can deter short-sighted management behavior and promote institutionalized and standardized business operations, thereby improving corporate governance [25, 26].

Following the principles of sustainable development, investors trust responsible and accountable enterprises, and management considers the long-term impact on enterprise value when making investment decisions Although enhancing ESG performance initially requires significant financial investments, it aligns with the cost-effectiveness principle by maximizing future economic benefits and sustainable development. Thus, governance and management are more likely to adopt sustainable development strategies that improve corporate governance and management capabilities. Enhanced internal governance levels significantly reduce audit risk by providing auditors with reliable information, allowing them to allocate limited audit resources to other critical areas and improve audit efficiency [27].

The concept of shared value emphasizes incorporating societal needs into a company's operations, creating business opportunities and competitive advantages by providing socially beneficial products and services, and generating both economic and social value. The introduction of ESG-related policies and increased competition in the business environment will drive a shift from "passive" to "active" disclosure of ESG performance. Companies will not abandon the principle of "profit maximization" but are more likely to adopt a "balancing"

strategy, considering their own economic interests alongside social benefits and seeking shared value. Many multinational corporations, such as GE, Nestle, Cisco, Coca-Cola, have become active proponents and practitioners of the shared value paradigm. Consequently, domestic listed companies in the early stages of ESG development will also explore new business models that strike a balance between economic and social benefits. Embracing this approach attracts investors, increases the confidence of debtors and the public in the company, and mitigates the impact of uncertain factors on business operations [28, 29].

According to the theory of social responsibility, actively assuming social and environmental responsibilities during the development process helps companies establish positive relationships with stakeholders such as regulators, suppliers, and customers. Research has indicated that companies with strong ESG performance face fewer financing constraints and reduced incentives for management to engage in earnings management, leading to lower financial risks and non-compliance risks. As a result, business risk is significantly reduced, providing auditors with more flexibility in formulating audit plans [30]. They can choose to employ fewer audit procedures while still effectively identifying and responding to the risks of material misstatement. This reduces the possibility of audit delays and improves overall audit efficiency. Based on the above analysis, the following research hypotheses can be proposed:

H1: Other things being equal, the better the firm's ESG performance, the more efficient the audit and the lower the audit delay.

Institutional investors, as important participants in the capital market, have information and professional advantages. Their supervision of the capital market can prevent listed companies from manipulating surpluses and improve the quality of governance and information transparency. Given their information and professional advantages, institutional investors play a pivotal role in the capital market. Through governance and synergies, institutional investors can improve the ESG performance of firms. firms with superior ESG performance are more likely to proactively disclose ESG-related information to the market, and even if institutional investors' shareholding is extremely low, they can significantly reduce information asymmetry, thus lowering the marginal benefit of their monitoring role. Based on the above analysis, the following research hypotheses can be formulated:

H2: The impact of external oversight is attenuated in firms with lower institutional investor ownership, and superior ESG performance is associated with increased audit efficiency and reduced audit delays, in contrast to firms with higher institutional investor ownership.

The "Big 4" accounting firms conduct audit engagements in many countries and have extensive experience in audit engagements related to environmental, social and corporate governance ratings. In addition, "Big 4" accounting firms have more robust internal control systems, regulations and more specialized talent than non-"Big 4" accounting firms." The "Big Four" accounting firms have stronger internal control systems, more robust regulations and more professionals, which reduces the likelihood of biased audit opinions and ensures the quality of information provided by the firms. The "Big Four" accounting firms are often regarded as models of high-quality audits. In addition, the capital market recognizes the impact of ESG ratings on audit risk, and the competence of auditors is enhanced for listed companies audited by "Big Four" accounting firms. Based on the above analysis, the following research hypotheses can be formulated:

H3: The effect of ESG performance on audit efficiency is more pronounced in firms that choose "Big 4" audits.

Under China's market system, enterprises of different ownership natures face different risk factors. Enterprises also differ in their environmental, social and governance performance. State-owned enterprises (SOEs) have a close relationship with the government and enjoy more social resources, and are therefore less constrained in terms of financing. On the contrary, NSOEs are more sensitive to national policies as they need to proactively improve their ESG performance in order to obtain favorable conditions or economic resources from policies. This implies that the marginal benefits arising from proactive social responsibility by non-SOEs will be higher than those of SOEs. Therefore, when non-state-owned enterprises enhance their ESG performance by improving their governance, it signals a subsequent reduction in their business risk. Based on the above analysis, the following research hypotheses can be formulated:

H4: The effect of firms' ESG performance on audit efficiency is more significant in non-state-owned firms.

## 4. Research design

### 4.1 Sample selection and data sources

In this study, we select A-share listed companies from 2015 to 2022 as our research sample. Based on which we further refined the sample by: excluding listed companies in the financial sector, omitting ST and *ST category firms, and removing instances with incomplete data. We also truncated the tails of all continuous variables at the 1% and 99% percentiles, ultimately yielding a panel dataset comprising 22,868 observations. The ESG data is sourced from the CSI ESG evaluation system provided by the Wind database, while the remaining financial data is obtained from the Cathay Pacific (CSMAR) database.

### 4.2 Definition of variables

**4.2.1 Explained variable.** Audit efficiency, also known as audit delay, is not only an indication of auditor efficiency, but also can be used as a proxy indicator of audit quality to some extent. Therefore, in this study, the number of days between the balance sheet date (December 31) and the audit report date is calculated and then the natural logarithm is applied to quantify the audit delay. It is understood that a larger value indicates a longer audit delay, which in turn indicates a less efficient audit.

**4.2.2 Explanatory variable.** ESG Performance. Currently, numerous ESG rating agencies operate in the market, including Bloomberg Data Terminal, Huazheng ESG Index, Shangdao Ronggreen, Runling Global, and Hexun.com CSR Report Rating Database. The CSI ESG ratings, crafted in alignment with national mainstream methodologies and practical experiences, and tailored to the nuances of China's capital market, are characterized by their rapid update frequency, extensive coverage, and high data reliability. Therefore, this study utilizes the CSI ESG ratings to assess the ESG performance of firms by categorizing the CSI ESG ratings into nine grades ranging from AAA to C, which are assigned values ranging from 9 to 1, respectively.

**4.2.3 Control variable.** This study selects firm size (Size), leverage ratio (Lev), ownership concentration as indicated by the proportion of shares held by the largest shareholder (Top1), CEO duality (Dual), institutional ownership (Inst), state ownership (Soe), and auditor reputation (Big4) as control variables. Additionally, year fixed effects (Year) and industry fixed effects (Indu) are included as controls. Detailed definitions of the main variables can be found in Table 1.

**Table 1. List of variable definitions.**

| Variable name | Variable name | Variable symbol | Description of variables |
|---|---|---|---|
| Explanatory variable | Audit efficiency | Aud | Number of days between the balance sheet date (December 31) and the date of the audit report and take the natural logarithm of that number |
| Explanatory variable | ESG performance | ESG | CSI ESG ratings from C to AAA are assigned as 1–9 respectively. |
| Control variable | Company size | Size | Natural logarithm of total assets at the end of the period |
| | Gearing | Lev | Ratio of total liabilities at the end of the period to total assets at the end of the period |
| | Shareholding ratio of the largest shareholder | Top1 | Number of shares held by the largest shareholder as a percentage of the total number of shares |
| | Two jobs in one | Dual | If the chairman and general manager are the same person, the value is 1, otherwise the value is 0. |
| | Institutional investors' shareholding | Anc | Proportion of A-share outstanding shares of the Company held by institutional investors such as funds, brokerages, QFIIs and insurance companies at the end of the year |
| | Nature of property rights | Soe | State-owned enterprises take the value of 1, otherwise it takes the value of 0 |
| | Firm Size | Big4 | If international "Big 4", the value is 1, otherwise the value is 0. |
| | Year (e.g. school year, fiscal year) | Year | Annual dummy variables |
| | Sector | Indu | Industry dummy variables, based on 2012 SEC industry classifications |
| Variables used in robustness testing | Corporate Growth | Growth | Measured by operating income growth rate |
| | Audit costs | Fee | Audit costs in logarithmic terms |
| | Type of audit opinion | Type | Value of 1 if standard unqualified opinion, otherwise value of 0 |

## 4.3 Research methodology

In this study, we used comprehensive panel data analysis, structural equation modeling (SEM), and a series of robustness tests to ensure the accuracy and reliability of the findings. First, the type of model applicable to the study data was determined by Hausman test. The ADF unit root test was performed on all time series variables to confirm data stability. The Breusch-Pagan/Cook-Weisberg test was used to diagnose heteroskedasticity in the model and adjusted with the help of robust standard errors. The Durbin-Watson test was used to assess the model serial correlation, and Pesaran's CD test to deal with cross-sectional dependence.

In addition, SEM was applied aiming to reveal causal relationships among variables, maximum likelihood estimation was used for parameter estimation, and model fit was assessed by CFI, RMSEA, and other metrics. The combined application of these methods and tests not only enhances the theoretical basis of the study, but also improves the broad applicability and credibility of the findings.

## 4.4 Modeling

The following model was designed for this study:

$$Aud_{i,\,t} = \alpha_0 + \alpha_1 ESG_{i,\,t} + \alpha_2 \text{Controls}_{i,\,t} + \Sigma Year + \Sigma Indu + \varepsilon_{i,\,t} \tag{1}$$

where Aud represents audit delay, $i$, $t$ denote listed companies and time, respectively, Controls represents a set of control variables, and the model also controls for year fixed effects (Year) and industry fixed effects (Indu) to minimize the impact of unobservables over time and factors related to industry characteristics.

**Table 2. Descriptive statistics of variables.**

| Variant | Sample size | Average value | (Statistics) Standard deviation | Minimum value | Upper quartile | Maximum values |
|---|---|---|---|---|---|---|
| Aud | 22868 | 4.601 | 0.178 | 3.989 | 4.663 | 4.787 |
| ESG | 22868 | 4.164 | 1.130 | 1.000 | 4.000 | 8.000 |
| Size | 22868 | 22.296 | 1.307 | 20.044 | 22.098 | 26.381 |
| Lev | 22868 | 0.409 | 0.198 | 0.059 | 0.400 | 0.883 |
| Top1 | 22868 | 0.341 | 0.144 | 0.094 | 0.319 | 0.740 |
| Dual | 22868 | 0.512 | 0.500 | 0.000 | 1.000 | 1.000 |
| Anc | 22868 | 0.436 | 0.251 | 0.003 | 0.450 | 0.919 |
| Soe | 22868 | 0.332 | 0.471 | 0.000 | 0.000 | 1.000 |
| Big4 | 22868 | 0.064 | 0.245 | 0.000 | 0.000 | 1.000 |

# 5. Empirical testing and analysis of results

## 5.1 Descriptive statistics

The results from Table 2, which show a maximum value of 4.79 and a minimum value of 3.99 for Audit Efficiency (Aud), suggest that the audit efficiency among the audited listed companies does not vary significantly. Conversely, the ESG performance exhibits a substantial variance among listed companies, with values ranging from a minimum of 1.00 to a maximum of 8.00, reflecting significant disparities in environmental (E), social (S), and corporate governance (G) performance. The descriptive statistical values for the remaining control variables appear to be more evenly distributed.

## 5.2 Correlation analysis

This study conducted a correlation analysis of the key variables, and the results from Table 3 reveal that the correlations between the variables are significant, providing preliminary validation for the study's hypotheses. Furthermore, the study performed Variance Inflation Factor (VIF) analysis on the key variables, and the findings indicate an absence of multicollinearity issues, suggesting that the selection of variables is appropriate.

## 5.3 Analysis of basic regression results

The empirical findings for the primary hypothesis are presented in columns (1)-(3) of Table 4. Specifically, column (1) shows the regression results excluding control variables and not

**Table 3. Table of correlation coefficients of main variables.**

|  | Aud | ESG | Size | Lev | Top1 | Dual | Anc | Soe | Big4 |
|---|---|---|---|---|---|---|---|---|---|
| Aud | 1 | -0.071*** | -0.040*** | 0.045*** | -0.071*** | 0.018*** | -0.112*** | -0.145*** | -0.115*** |
| ESG | -0.051*** | 1 | 0.157*** | -0.058*** | 0.102*** | 0.021*** | 0.097*** | 0.068*** | 0.114*** |
| Size | -0.039*** | 0.195*** | 1 | 0.521*** | 0.137*** | -0.041*** | 0.428*** | 0.381*** | 0.271*** |
| Lev | 0.033*** | -0.068*** | 0.522*** | 1 | 0.031*** | -0.020*** | 0.192*** | 0.281*** | 0.113*** |
| Top1 | -0.053*** | 0.111*** | 0.189*** | 0.038*** | 1 | -0.024*** | 0.475*** | 0.210*** | 0.136*** |
| Dual | 0.021*** | 0.022*** | -0.030*** | -0.017** | -0.018*** | 1 | -0.259*** | 0.104*** | -0.053*** |
| Anc | -0.098*** | 0.101*** | 0.453*** | 0.194*** | 0.482*** | -0.264*** | 1 | 0.421*** | 0.255*** |
| Soe | -0.115*** | 0.073*** | 0.396*** | 0.288*** | 0.218*** | 0.104*** | 0.425*** | 1 | 0.132*** |
| Big4 | -0.099*** | 0.113*** | 0.344*** | 0.111*** | 0.152*** | -0.053*** | 0.259*** | 0.132*** | 1 |

*** p<0.01

** p<0.05

* p<0.1

**Table 4. Corporate ESG performance and audit efficiency.**

| Variant | (1) Aud | (2) Aud | (3) Aud |
|---|---|---|---|
| ESG | -0.010*** | -0.005*** | -0.007*** |
|  | (-9.83) | (-4.43) | (-6.67) |
| Size |  | 0.004*** | 0.004*** |
|  |  | (2.94) | (3.12) |
| Lev |  | 0.062*** | 0.055*** |
|  |  | (8.74) | (7.67) |
| Top1 |  | 0.003 | 0.008 |
|  |  | (0.34) | (0.89) |
| Dual |  | 0.007*** | 0.009*** |
|  |  | (2.74) | (3.57) |
| Anc |  | -0.031*** | -0.021*** |
|  |  | (-4.84) | (-3.39) |
| Soe |  | -0.043*** | -0.039*** |
|  |  | (-14.66) | (-13.84) |
| Big4 |  | -0.062*** | -0.060*** |
|  |  | (-12.09) | (-12.22) |
| Constant | 4.593*** | 4.545*** | 4.497*** |
|  | (396.54) | (181.14) | (168.82) |
| N | 22,868 | 22,868 | 22,868 |
| R-squared | 0.095 | 0.029 | 0.114 |
| Indu FE | YES | NO | YES |
| Year FE | YES | NO | YES |

*** $p<0.01$

** $p<0.05$

* $p<0.1$

accounting for year and industry effects, column (2) represents the regression outcomes after incorporating control variables but still without adjusting for year and industry effects, and column (3) details the regression analysis that includes control variables and additionally adjusts for industry and year effects. All regression models are significant at the 1% level, demonstrating a substantial correlation between firms' ESG performance and their audit efficiency, namely, better ESG performance in firms indeed leads to reduced audit delays and enhanced audit efficiency.

## 5.4 SME analysis

Considering the insufficient explanatory power of the existing model, structural equation modeling was continued to be applied to validate the variables and the results are shown in Table 5 and Fig 1.

As can be seen from the above table, from the results of structural equation modeling (SEM) analysis, the estimated value of ESG is -0.555, the standard error is 0.097, the z-value is -5.728, and the P-value is 0, which indicates that ESG has a significant negative correlation with Aud, i.e., an increase in the indicator of ESG may lead to a decrease in the value of Aud. the estimated value of Size is 0.218, the standard error is 0.106, z-value is 2.06, p-value is 0.039, showing that Size has a positive but relatively small and statistically significant effect on Aud. the estimated value of Lev is 6.146, standard error is 0.612, z-value is 10.045, p-value is 0,

**Table 5. Results of structural equation modeling analysis.**

| Variables | Estimate | Std.Err | z-value | P(>|z|) | Std.lv | Std.all |
|---|---|---|---|---|---|---|
| ESG | -0.555 | 0.097 | -5.728 | 0 | -0.555 | -0.038 |
| Size | 0.218 | 0.106 | 2.06 | 0.039 | 0.218 | 0.018 |
| Lev | 6.146 | 0.612 | 10.045 | 0 | 6.146 | 0.078 |
| Top1 | 0 | 0.008 | 0.036 | 0.972 | 0 | 0 |
| Dual | 0.909 | 0.235 | 3.864 | 0 | 0.909 | 0.026 |
| Anc | -0.034 | 0.005 | -6.463 | 0 | -0.034 | -0.055 |
| Soe | -4.109 | 0.257 | -15.974 | 0 | -4.109 | -0.121 |
| Big4 | -6.238 | 0.451 | -13.844 | 0 | -6.238 | -0.096 |

indicating that Lev has a very significant positive correlation with Aud, which is the most significant of all the variables that affect Aud. Aud. The estimate of Top1 is 0, standard error is 0.008, z-value is 0.036, and p-value is 0.972, indicating that Top1 has no significant effect on Aud. The estimate of Dual is 0.909, standard error is 0.235, z-value is 3.864, and p-value is 0, showing that Dual has a positive correlation with Aud and the influence is significant. Anc has an estimate of -0.034, a standard error of 0.005, a z-value of -6.463, and a p-value of 0, showing that Anc has a significant negative correlation with Aud, albeit with a relatively small influence. Soe has an estimate of -4.109, a standard error of 0.257, and a z-value of -15.974, with a p-value of 0, showing that Soe has a very strong negative correlation with Aud. big4 has an estimated value of -6.238, a standard error of 0.451, a z-value of -13.844, and a p-value of 0, indicating that big4 has a strong negative correlation with Aud.

In summary, Lev, Soe and Big4 are the most influential variables on Aud, with Lev having a strong positive impact on Aud, while Soe and Big4 have a strong negative impact. ESG, Size, Dual and Anc are statistically significant, although their influence is relatively small. In particular, ESG and Anc show a significant negative correlation. Top1 has a non-significant effect on Aud.

In terms of the fitting indices, the ratio of the chi-square value and the degree of freedom are higher than the judgment standard, which is due to the excessive number of samples in this case, resulting in the chi-square value being too large. All other fitting term indices meet the requirements of judgment criteria, indicating that the variables in the structural equation model are fitted better, and the results are shown in Table 6.

### 5.5 Further studies

**5.5.1 Heterogeneity analysis of institutional investor holdings.** The analysis segregates firms into two cohorts based on whether the share of institutional investor ownership exceeds

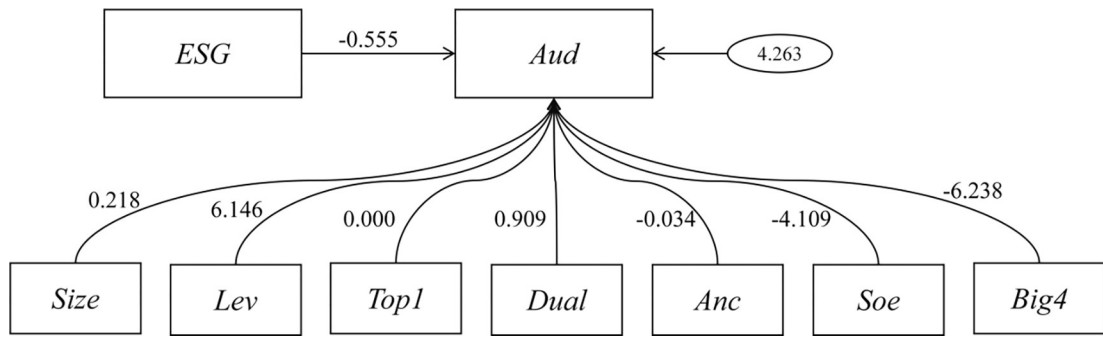

**Fig 1. Structural equation modeling of factors affecting audit efficiency.**

**Table 6. Fitting indexes of structural equation model.**

| Model Fit Indicators | in the end | standard of judgment |
|---|---|---|
| absolute fit index (AFI) | | |
| Goodness-of-fit index (GFI) | 0.987 | Range of values 0~1, >0.9 for good fit |
| Adjusted Goodness of Fit Index (AGFI) | 0.924 | Range of values 0~1, >0.9 for good fit |
| Root Mean Square Error of Approximation (RMSEA) | 0.003 | Less than 0.05 indicates a good fit |
| Charted values ($\chi^{2/v}$) | 2.742 | <3 indicates good fit |
| Relative fit index | | |
| Normative Fit Index (NFI) | 0.945 | Range of values 0~1, >0.9 for good fit |
| Comparative Fit Index (CFI) | 0.956 | Range of values 0~1, >0.9 for good fit |

the industry-annual median: one with a high proportion and another with a low proportion of institutional investors. Subsequently, model (1) is evaluated for each group, with findings presented in columns (1) and (2) of Table 7. Both groups yield results significant at the 1% level,

**Table 7. Results of heterogeneity test.**

| Variant | Institutional investor holdings | | Audit quality | | Nature of property rights | |
|---|---|---|---|---|---|---|
| | your (honorific) | lower (one's head) | "The Big Four" | "Not the Big Four" | state enterprise | non-state enterprise |
| | (1) | (2) | (3) | (4) | (5) | (6) |
| | Aud | Aud | Aud | Aud | Aud | Aud |
| ESG | -0.005*** | -0.008*** | -0.021*** | -0.006*** | -0.006*** | -0.008*** |
| | (-3.51) | (-5.53) | (-5.49) | (-5.60) | (-3.16) | (-6.42) |
| Size | 0.003** | 0.005*** | -0.006 | 0.005*** | 0.011*** | 0.002 |
| | (2.00) | (2.71) | (-1.58) | (4.13) | (5.57) | (0.95) |
| Lev | 0.056*** | 0.049*** | 0.027 | 0.054*** | 0.065*** | 0.045*** |
| | (5.44) | (4.92) | (1.04) | (7.29) | (5.69) | (4.91) |
| Top1 | 0.044*** | -0.028** | 0.016 | 0.010 | 0.041** | -0.005 |
| | (3.42) | (-2.03) | (0.56) | (1.08) | (2.50) | (-0.48) |
| Dual | 0.013*** | 0.004 | 0.039*** | 0.007*** | 0.018*** | 0.001 |
| | (3.65) | (1.22) | (5.01) | (2.71) | (4.85) | (0.41) |
| Anc | -0.066*** | -0.017 | -0.038 | -0.025*** | -0.073*** | -0.021*** |
| | (-4.55) | (-1.19) | (-1.54) | (-3.83) | (-4.72) | (-2.91) |
| Soe | -0.046*** | -0.032*** | -0.076*** | -0.036*** | | |
| | (-12.57) | (-6.56) | (-8.57) | (-12.07) | | |
| Big4 | -0.060*** | -0.053*** | | | -0.085*** | -0.031*** |
| | (-10.99) | (-4.39) | | | (-13.43) | (-4.26) |
| Constant | 4.548*** | 4.451*** | 4.652*** | 4.461*** | 4.360*** | 4.524*** |
| | (128.74) | (103.61) | (50.83) | (158.02) | (110.01) | (122.42) |
| N | 11,390 | 11,478 | 1,471 | 21,397 | 7,583 | 15,285 |
| R-squared | 0.120 | 0.115 | 0.235 | 0.105 | 0.143 | 0.099 |
| Indu FE | YES | YES | YES | YES | YES | YES |
| Year FE | YES | YES | YES | YES | YES | YES |
| P-value | 0.084 | | 0.000 | | 0.001 | |

Note: (1)

*** p<0.01

** p<0.05

* p<0.1; (2) P-values for the test of coefficient intergroup differences in the analysis of heterogeneity were calculated using the Fisher's Combined Test (1,000 samples).

prompting the execution of a Fisher's exact test, which yields a p-value of 0.084—indicating significance at the 10% threshold. This suggests that the effect of corporate environmental, social and governance performance on audit efficiency is more pronounced in firms with lower institutional investor ownership, which confirms hypothesis H2.

**5.5.2 Heterogeneity analysis of audit quality.** The listed companies in this study are categorized into "Big 4" and "Non-Big 4" and model (1) is evaluated in these two categories and the results are shown in columns (3) and (4) of Table 7. All regression results show significance at the 1% level, thus requiring a Fisher's portfolio test. The resulting p-value of 0.000 indicates significance at the 1% level, confirming that the contribution of corporate ESG performance to audit efficiency is greater among firms choosing the "Big 4" audits, thus testing hypothesis H3.

**5.5.3 Heterogeneity analysis of the nature of property rights.** In this study, listed companies are categorized into two types of state-owned and non-state-owned enterprises, and model (1) is evaluated in these two types of enterprises, and the results are shown in columns (5) and (6) of Table 7. All regression results show significance at the 1% level, thus requiring a Fisher's portfolio test. The obtained P-value of 0.000 indicates significance at the 1% level, confirming that the contribution of corporate ESG performance to audit efficiency is more significant in non-state-owned firms, thus testing hypothesis H4.

## 6. Robustness tests

### 6.1 One-period lag test

To address potential endogeneity issues, we apply a one-period lag to the ESG data to examine the continued impact of firms' ESG performance on audit delays. The regression results presented in Column (1) of Table 6 indicate that firms' ESG performance and audit efficiency remain statistically significant at the 1% level, thereby corroborating the primary research findings of the article.

### 6.2 Heckman two-stage

To construct the first stage model, we introduce a dummy variable representing the level of firms' ESG as the dependent variable. This dummy variable takes a value of 1 if the firm's ESG level surpasses the annual-industry average, and 0 otherwise. Additionally, we employ a Probit model incorporating the control variables from model (1) to estimate the inverse Mills coefficient (IMR). The specific model setup is outlined below.

$$
\begin{aligned}
\text{Probit}&(\text{ESGi, t}) \\
&= \beta_0 + \beta_1 \, \text{Size}_{i,t} + \beta_2 \, \text{Lev}_{i,t} + \beta_3 \, \text{Top1}_{i,t} + \beta_4 \, \text{Dual}_{i,t} + \beta_5 \, \text{Anc}_{i,t} + \beta_6 \, \text{Soe}_{i,t} \\
&\quad + \beta_7 \, \text{Dual}_{i,t} + \beta_7 \, \text{Big4}_{i,t} + \Sigma \text{Year} + \Sigma \text{Indu} + \varepsilon_{i,t} \quad\quad\quad (2)
\end{aligned}
$$

Following the estimation of the Inverse Mills Ratio (IMR) from the aforementioned model, it is incorporated into the primary regression model and re-estimated. The regression findings, presented in column (2) of Table 8, exhibit statistical significance at the 5% level, thereby reinforcing the principal findings of this paper.

### 6.3 Double clustering adjustment

To effectively mitigate the impact of heteroskedasticity and autocorrelation issues on the statistical outcomes, this study employs the double clustering adjustment method to address the clustering of standard errors at both the individual and industry levels. The regression outcomes, presented in column (3) of Table 8, remain statistically significant, thereby affirming the validity of the research findings even after employing the double clustering adjustment.

**Table 8. Robustness test.**

| Variant | (1) Aud | (2) Aud | (3) Aud | (4) Aud |
|---|---|---|---|---|
| ESG | -0.007*** | -0.004*** | -0.007*** | -0.005*** |
| | (-5.67) | (-4.03) | (-6.70) | (-4.76) |
| Size | 0.004* | 0.003** | 0.027*** | -0.005*** |
| | (1.79) | (2.37) | (2.74) | (-3.36) |
| Lev | 0.055*** | 0.059*** | -0.066 | 0.040*** |
| | (4.06) | (8.31) | (-1.29) | (5.55) |
| Top1 | 0.008 | 0.007 | 0.056** | 0.017* |
| | (0.41) | (0.74) | (2.54) | (1.88) |
| Dual | 0.009* | 0.008*** | 0.013*** | 0.008*** |
| | (1.92) | (3.46) | (4.30) | (3.45) |
| Anc | -0.021 | -0.021*** | -0.035*** | -0.026*** |
| | (-1.48) | (-3.36) | (-4.08) | (-4.10) |
| Soe | -0.039*** | -0.039*** | -0.035*** | -0.036*** |
| | (-6.33) | (-13.85) | (-10.20) | (-12.51) |
| Big4 | -0.060*** | -0.060*** | -0.044*** | -0.077*** |
| | (-8.98) | (-12.28) | (-5.27) | (-14.95) |
| Income | | | | 0.000 |
| | | | | (0.53) |
| Fee | | | | 0.027*** |
| | | | | (10.02) |
| Type | | | | -0.078*** |
| | | | | (-10.46) |
| IMR | | | -0.339** | |
| | | | (-2.39) | |
| Constant | 4.498*** | 4.506*** | 4.454*** | 4.400*** |
| | (106.80) | (169.10) | (137.46) | (140.58) |
| N | 22,832 | 22,832 | 22,828 | 22,832 |
| R-squared | 0.114 | 0.113 | 0.114 | 0.123 |
| Indu FE | YES | YES | YES | YES |
| Year FE | YES | YES | YES | YES |

*** p<0.01

** p<0.05

* p<0.1

## 6.4 Further control of other variables

Given that other factors may potentially influence the association between firms' ESG level and audit delay and consequently impact the primary test outcomes, this study employs additional control variables, including firm growth (Growth, measured by the revenue growth rate), audit fee (Fee, represented in the logarithmic form), and audit opinion type (Type, taking a value of 1 for a standard unqualified opinion and 0 otherwise). The regression results, presented in column (4) of Table 8, confirm that the principal findings hold even after controlling for these additional variables.

## 7. Talk over

This study demonstrates a significant positive association between corporate ESG performance and audit efficiency, especially among firms with low institutional investor ownership, firms

audited by Big 4 accounting firms, and non-state-owned firms. These findings not only enrich the existing literature on ESG and audit efficiency, but also provide practical guidance for managers and policymakers, emphasizing the importance of enhancing corporate ESG performance to improve audit efficiency.

First, our results echo stakeholder and social responsibility theories by emphasizing the important role of firms' performance in environmental, social responsibility and governance in reducing audit risk and improving information transparency. By comparing with previous studies, such as the findings of Giese et al. (2019), Kim and Li (2021), and Zhao et al. (2023) [31–33], our results further confirm that good ESG performance can serve as an effective tool for corporate risk management and thus improve audit efficiency.

Second, this study reveals differences in the impact of ESG performance on audit efficiency among different types of firms through a careful heterogeneity analysis. This point highlights the need for strategies to enhance ESG performance to be targeted under different governance structures and market environments. In particular, the more significant results for non-state-owned firms and non-Big 4 audit clients suggest that ESG investments and improvements are particularly critical for enhancing audit efficiency in these environments.

Finally, these findings provide important insights for practice. For managers, actively improving a firm's ESG performance not only enhances the company's image, but also directly impacts the efficiency and quality of the audit process. For auditors, this emphasizes the need for greater consideration of firms' ESG performance as part of audit planning during the audit process. Policymakers, for their part, should consider how to encourage companies to improve their ESG performance through policies that promote transparency and efficiency in the market as a whole.

## 8. Research conclusions and policy recommendations

This study utilizes a sample of A-share listed companies from 2015–2022 to investigate the effect of corporate ESG performance on audit efficiency from the perspective of audit delay. The results demonstrate that firms exhibiting good ESG performance experience a lower likelihood of audit delay and higher audit efficiency. These findings hold even after conducting tests with lagged one-period, Heckman's two-stage, double clustering adjustment, and controlling for additional variables. Further analysis indicates that the contribution of firms' ESG performance to audit efficiency is more significant in firms with lower institutional investor ownership, firms audited by the Big 4, and non-state-owned firms.

The present study enriches the existing literature on corporate ESG performance and auditing efficiency, which enhances the public's overall understanding of the impact of corporate ESG performance on auditing. Given the practical significance of ESG construction in promoting economic and social development and achieving the goal of "double carbon," this paper proposes the following policy recommendations: Firstly, audit committees play a crucial role in overseeing the fulfillment of corporate sustainable development strategies. However, most audit committees are not well-versed in ESG-related laws and guidelines, and their understanding of sustainable development is at a basic level. Therefore, it is imperative to provide training for audit committee directors to enhance their competence. Secondly, China is still in the exploratory stage of ESG construction and has yet to establish complete ESG standards and disclosure systems. In addition, the capital market remains skeptical of the information published by listed companies. Therefore, regulators should accelerate the development of an ESG evaluation system suitable for China's actual situation, and mandate independent or special audits of corporate ESG-related reports to enhance the credibility of the information. Thirdly, accounting firms should introduce and cultivate ESG audit talents and assign

personnel with professional competence to conduct audits of listed companies, while identifying risk factors in ESG reports. They should also establish and improve the specific operational processes and substantive execution procedures related to ESG business.

Although this study provides empirical evidence on the relationship between corporate ESG performance and audit efficiency, there are some limitations. First, the sample is limited to A-share listed companies, which may not be fully representative of other markets or countries. Second, this study mainly relies on publicly disclosed data, which may suffer from incomplete information. Finally, there is a certain degree of subjectivity in the ESG evaluation system, and there may be differences in the evaluation results of different organizations. Future research can further explore the relationship between ESG performance and audit efficiency in different industry or country contexts to verify the generalizability of the results of this study. Meanwhile, the use of more dimensions of data, such as internal audit quality, corporate governance structure and other factors, can be considered to deeply analyze their impact on audit efficiency. In addition, with the continuous improvement and harmonization of ESG evaluation standards, future research can explore the differences in the impact of corporate ESG performance on audit efficiency under different ESG evaluation systems.

## Supporting information

**S1 Dataset.**
(XLSX)

## Author Contributions

**Conceptualization:** Li Zhang, Caixia Guo.

**Data curation:** Li Zhang, Caixia Guo.

**Formal analysis:** Li Zhang, Caixia Guo.

**Funding acquisition:** Li Zhang.

**Investigation:** Li Zhang, Caixia Guo.

**Methodology:** Li Zhang, Caixia Guo.

**Validation:** Li Zhang, Caixia Guo.

**Visualization:** Caixia Guo.

**Writing – original draft:** Li Zhang, Caixia Guo.

**Writing – review & editing:** Li Zhang, Caixia Guo.

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
