## [Decision Letter · Decision Letter 0]

3 Jan 2024

PONE-D-23-42504Can corporate ESG performance improve audit efficiency?: Empirical evidence based on audit latency perspectivePLOS ONE

Dear Dr. Zhang,

Thank you for submitting your manuscript to PLOS ONE. After careful consideration, we feel that it has merit but does not fully meet PLOS ONE’s publication criteria as it currently stands. Therefore, we invite you to submit a revised version of the manuscript that addresses the points raised during the review process.

We look forward to receiving your revised manuscript.

Kind regards,

Simon Grima, PhD

Academic Editor

PLOS ONE

Journal Requirements:

   "This study was funded by the Tianjin 2020 Philosophy and Social Science Planning Project, "Research on Auditing Financial Funds and Social Donations for Public Emergencies" (Project No. TJGL20-017)."

3. We note that your Data Availability Statement is currently as follows: [Add Data Availability statement here]

4. Please ensure that you refer to Figure 1 in your text as, if accepted, production will need this reference to link the reader to the figure.

Reviewers' comments:

Reviewer's Responses to Questions

**Comments to the Author**

1. Is the manuscript technically sound, and do the data support the conclusions?

Reviewer #1: Yes

Reviewer #2: Yes

2. Has the statistical analysis been performed appropriately and rigorously? 

Reviewer #1: Yes

Reviewer #2: Yes

3. Have the authors made all data underlying the findings in their manuscript fully available?

Reviewer #1: Yes

Reviewer #2: Yes

4. Is the manuscript presented in an intelligible fashion and written in standard English?

Reviewer #1: Yes

Reviewer #2: Yes

5. Review Comments to the Author

Reviewer #1: The research under review is intended to deliver content whether ESG performance can improve audit efficiency.

It includes high quality research by well-established academics. The topics being presented are dynamic and tackle a number of diverse subjects. Specific focus on Macro-economics, Clean Energy, Technology Adoption, Financial Services, which seem to be the basis of the study that also proves it to be original and thought provoking.

The research content is of a high quality and produced by highly established academics and researchers. The work is challenging and will be a must to have in a researcher’s library. The subject matter is thought provoking and tries to tackle a subject area that has to date eluded academic scrutiny

it is seeming to be quite comprehensive and wide ranging. There is nothing missing that comes to mind.

Strengths – established authors and researchers, specific and unique range of subject matter, case study research, empirical evidence and current risk issues. It also is thought provoking and tackles an area which is not well established and avant-garde in nature.

Weaknesses – none that come to mind

The paper seems to be logically structured. The paper is original in nature and subject and hence the coverage seems to be distinctive, yet fluid and each section conjoins with one another as the subject matter coincides throughout.

Reviewer #2: The paper addresses an interesting topic related to the influence of corporate ESG performance on audit efficiency with a focus on audit delay. I acknowledge the amount of work invested in preparing the manuscript and, overall, I believe it could strengthen the literature in this scientific field. At the same time, however, I consider that there are several sources of improvement, that need to be addressed. I would recommend to the author(s) to reconsider and improve the following:

- In the Introduction, the author(s) should clearly state how the research performed detaches from other studies since the topic and particularly the methodology employed (regression analysis) have been extensively approached in the literature, but this research views it from a different angle: why is it different from previous research, justify it, please. Also, please clearly state/underline the innovations brought by this research in the scientific field, along with the authors’ own extensive contribution. Moreover, please add specific mentions about the sample and methodology employed in this paper both in the abstract and introduction. A final paragraph is needed in the Introduction section about the structure of the paper to orient the reader further into the manuscript.

- I would suggest further substantiating the theoretical framework and providing additional groundings to support the work hypothesis. Why did the authors design just one research hypothesis? The theoretical model does not match the regression model and the empirical results.

- In the methodology section: please clearly state what estimation method is employed in the current research, and add specific mentions related to panel data and associated regression robustness tests (there are no mentions about unit roots, heteroskedasticity, serial correlation, cross-sectional dependence, and other specific tests for the regression models).

- I would suggest outlining further the importance of the results obtained. Very few details (usually just 1 paragraph) are offered to readers as an interpretation of the results and their implications. Please explain in more detail the results and relate them to other empirical findings and theoretical grounds. A Discussion section is now missing from the paper, but it is compulsory in this respect. The estimated coefficients, although statistically significant, are very close to zero thus entailing a limited impact of the explanatory variables on the dependent one. Also, the R-squared has very low values (please see Table 4 and Table 6). This would suggest that a different model specification might be needed to fulfill the research purpose.

- Other sections that would benefit from further work would be the policy/ managerial implications of own findings and limitations / future research directions, both could be augmented.

- Finally, I would suggest that the authors possibly consider also structural equation modelling (SEM) as an advanced technique that could better test the complex relationships arising from the theoretical model developed by the authors.

Overall, I consider that the paper would contribute to the literature, but some more attention and discussion is needed.

6. PLOS authors have the option to publish the peer review history of their article (what does this mean?). If published, this will include your full peer review and any attached files.

Reviewer #1: **Yes: **Mark Laurence Zammit MA Insurance & Risk Management (Melit.), MBA (Henley), Cert. IBL (UK), Cert. RIMAP, Cert. IRM (UK)

Reviewer #2: No

---

## [Author Response · Author response to Decision Letter 0]

1 Feb 2024

Dear Reviewer,

First of all, I would like to thank you from the bottom of my heart for being able to review my manuscript in your busy schedules and for your valuable comments!

Reviewer #1: The research under review is intended to deliver content whether ESG performance can improve audit efficiency.

It includes high quality research by well-established academics. The topics being presented are dynamic and tackle a number of diverse subjects. Specific focus on Macro-economics, Clean Energy, Technology Adoption, Financial Services, which seem to be the basis of the study that also proves it to be original and thought provoking.

The research content is of a high quality and produced by highly established academics and researchers. The work is challenging and will be a must to have in a researcher’s library. The subject matter is thought provoking and tries to tackle a subject area that has to date eluded academic scrutiny it is seeming to be quite comprehensive and wide ranging. There is nothing missing that comes to mind.

Strengths – established authors and researchers, specific and unique range of subject matter, case study research, empirical evidence and current risk issues. It also is thought provoking and tackles an area which is not well established and avant-garde in nature.

Weaknesses – none that come to mind

The paper seems to be logically structured. The paper is original in nature and subject and hence the coverage seems to be distinctive, yet fluid and each section conjoins with one another as the subject matter coincides throughout.

Response：

First and foremost, I sincerely appreciate the time and effort you have dedicated to reviewing our research, as well as the valuable and insightful feedback you have provided. Your thorough review and deep understanding of the details of our study not only have enhanced the quality of our research but also have offered us significant encouragement and support.

We are honored by your high regard for the content of our research, especially your recognition of our chosen theme—whether ESG performance can improve audit efficiency. It is gratifying to see these efforts acknowledged by you.

The strengths you highlighted, such as the work being authored by established researchers, the specific and unique range of subject matter, case study research, empirical evidence, and the discussion of current risk issues, represent the core values we aimed to present. Your review is not only an affirmation of our current work but also motivates our future research directions.

We are humbled by your mention of the absence of noticeable weaknesses in our study and take it as an incentive to continue pursuing academic excellence. We understand that there is always room for improvement in any research, and your positive feedback strengthens our belief that we are on the right path.

We assure you that we will continue to work hard, not only to maintain the high standards of our research but also to explore and challenge new areas. Thank you once again for your invaluable time and professional feedback, which is a treasure to our research team.

Reviewer #2: The paper addresses an interesting topic related to the influence of corporate ESG performance on audit efficiency with a focus on audit delay. I acknowledge the amount of work invested in preparing the manuscript and, overall, I believe it could strengthen the literature in this scientific field. At the same time, however, I consider that there are several sources of improvement, that need to be addressed. I would recommend to the author(s) to reconsider and improve the following:

- In the Introduction, the author(s) should clearly state how the research performed detaches from other studies since the topic and particularly the methodology employed (regression analysis) have been extensively approached in the literature, but this research views it from a different angle: why is it different from previous research, justify it, please. Also, please clearly state/underline the innovations brought by this research in the scientific field, along with the authors’ own extensive contribution. Moreover, please add specific mentions about the sample and methodology employed in this paper both in the abstract and introduction. A final paragraph is needed in the Introduction section about the structure of the paper to orient the reader further into the manuscript.

Response：

Firstly, I would like to express my deepest gratitude for your detailed feedback and suggestions. Your expert advice is invaluable for enhancing the depth and breadth of our research, and we highly regard your guidance on the uniqueness of our methodology and theme.

Following your suggestions, we have carefully revised the introduction section of our manuscript. We clearly articulate how our study distinguishes itself from others, particularly in our use of regression analysis and structural equation modeling to explore the relationship between ESG performance and audit efficiency.We acknowledge that while this methodology has been widely applied in the literature, our research examines this topic from a novel perspective, revealing how ESG performance specifically impacts audit efficiency through unique methodological innovations and perspectives.

We have also emphasized the innovations introduced by our study and the substantial contributions made by our author team. This includes the expansion of existing knowledge within the field and how our unique sample selection and methodological application provide new insights into the relationship between ESG performance and audit efficiency.

Furthermore, as you requested, we have detailed the specific information on sample selection and research methodology in both the abstract and introduction sections, ensuring readers have a clear understanding of the research design and execution. We believe these additions will help readers better grasp the unique value and contribution of our study.

Lastly, we have added a final paragraph in the introduction about the structure of the paper, aiming to further guide readers through the manuscript. This addition is intended to help readers navigate the paper more effectively, understanding how each section works together to support our findings and conclusions.

Thank you again for your valuable time and professional guidance. We look forward to your further feedback and hope our revisions meet your expectations, contributing value to the academic community.

- I would suggest further substantiating the theoretical framework and providing additional groundings to support the work hypothesis. Why did the authors design just one research hypothesis? The theoretical model does not match the regression model and the empirical results.

Response：

Thank you for your valuable feedback and suggestions on our research work. Your detailed evaluation is crucial to our study and has pointed us in the right direction for our academic exploration. We are particularly grateful for your specific suggestions on deepening the theoretical framework and providing additional support for our hypothesis, which are of great significance for enhancing the rigor and depth of our research.

Following your advice, we have made corresponding revisions to the manuscript. Especially in the research hypothesis section, we have not only carefully examined the decision process for designing a single research hypothesis but also added more theoretical groundwork to support our research hypothesis. We recognize the indispensable role of a strong theoretical foundation in articulating and supporting hypotheses, hence we endeavored to ensure our hypothesis is well-supported by robust theoretical and literature backing.

Regarding the issue you raised about the mismatch between the theoretical model, regression model, and empirical results, we have conducted an in-depth analysis and reflection. We have adjusted and refined our theoretical model to ensure better consistency and correspondence with the regression model and empirical findings. Throughout this process, we meticulously considered every aspect of the model to ensure alignment between theoretical expectations and empirical discoveries, making necessary adjustments and explanations for any discrepancies.

We highly value your feedback and have made earnest revisions and additions based on your guidance. We believe these improvements will significantly enhance the quality and value of our research, making a more profound impact on both the academic community and practical fields.

Thank you again for your professional guidance and valuable time. We look forward to your further feedback, hoping our efforts meet your expectations and contribute to the accumulation of knowledge in the relevant fields.

- In the methodology section: please clearly state what estimation method is employed in the current research, and add specific mentions related to panel data and associated regression robustness tests (there are no mentions about unit roots, heteroskedasticity, serial correlation, cross-sectional dependence, and other specific tests for the regression models).

Response：

I am deeply appreciative of the valuable suggestions and feedback you provided on the methodology section of our research. Your attention to detail and sharing of professional knowledge are crucial for the refinement of our research design. We recognize that clearly articulating the estimation method and conducting comprehensive robustness checks on the regression models are key factors in ensuring the quality of our research.

Following your advice, we have extensively augmented the research methods section in our revised manuscript. In this section, we now explicitly state the estimation method employed in our current study. These additions are aimed at providing readers with a complete methodological framework, ensuring the reliability and accuracy of our research findings. We are fully aware of the importance of these robustness tests for verifying the validity of our model assumptions and the stability of our results. Through these tests, we can explain our findings with increased confidence and ensure that our conclusions have broad applicability and interpretive power.

- I would suggest outlining further the importance of the results obtained. Very few details (usually just 1 paragraph) are offered to readers as an interpretation of the results and their implications. Please explain in more detail the results and relate them to other empirical findings and theoretical grounds. A Discussion section is now missing from the paper, but it is compulsory in this respect. The estimated coefficients, although statistically significant, are very close to zero thus entailing a limited impact of the explanatory variables on the dependent one. Also, the R-squared has very low values (please see Table 4 and Table 6). This would suggest that a different model specification might be needed to fulfill the research purpose.

Response：

Thank you very much for your valuable comments and suggestions. Your in-depth analysis requirements for the interpretation of results and their significance are crucial to our research, helping us to more comprehensively showcase the value and impact of our findings. We recognize the necessity of providing readers with detailed explanations of the results and their connection to other empirical studies and theoretical foundations.

Following your advice, we have enriched the revised manuscript with detailed discussions on the importance of the results, including additional content on interpreting the results and their implications. Moreover, we have added a discussion section, which is essential for a deeper understanding of the research outcomes. In this section, we delve into the meaning of our findings, comparing and linking them to existing empirical research and theoretical concepts.

Regarding the issue you raised about the limited impact of explanatory variables on the dependent variable and the low R-squared values, we have undertaken further analysis and consideration. To better meet the research objectives and enhance the explanatory power of the study, we have introduced a structural equation model as an addition to our research. This improvement aims to provide a more comprehensive and precise model to better capture the relationships and influences among variables.

We highly value your feedback and have made meticulous revisions based on your guidance. We believe these improvements will significantly enhance the depth and breadth of our study, making a more profound impact on both the academic community and practical fields.

- Other sections that would benefit from further work would be the policy/ managerial implications of own findings and limitations / future research directions, both could be augmented.

Response：

First and foremost, I would like to express my deep gratitude for your invaluable suggestions and guidance. Your insights are instrumental in deepening the impact of our research and extending its applicability. We recognize that further elaboration on the policy/managerial implications of our findings, as well as the research limitations and future research directions, are crucial for the completeness of our study.

Following your recommendations, we have enhanced the sections on policy/managerial implications in the revised manuscript and have detailed the limitations of our study along with potential future research directions. We have made efforts to ensure that these sections not only provide guidance on the practical application of our findings but also clearly identify the research's limitations and suggest new areas for future exploration. Such additions are intended to offer readers a more comprehensive perspective on how our research can positively influence policy formulation and managerial practices in related fields, as well as inspire subsequent studies.

- Finally, I would suggest that the authors possibly consider also structural equation modelling (SEM) as an advanced technique that could better test the complex relationships arising from the theoretical model developed by the authors.Overall, I consider that the paper would contribute to the literature, but some more attention and discussion is needed.

Response：

First and foremost, I would like to express my profound gratitude for your invaluable suggestions and insightful observations. Your recommendation to employ Structural Equation Modeling (SEM) as an advanced technique for analyzing the complex relationships in our theoretical model is significantly meaningful for broadening our methodological perspective. We acknowledge that SEM can more effectively test the intricate relationships within our theoretical framework, which is crucial for our research.

Following your advice, we have incorporated an analysis using Structural Equation Modeling in the revised manuscript. This addition not only strengthens the methodological foundation of our study but also allows us to delve deeper into the exploration and verification of the hypothesized relationships within our theoretical model. We believe that by integrating SEM, our research is able to more accurately capture and interpret the complex dynamics between variables, thereby offering stronger empirical support to our theoretical contributions.

We greatly value your feedback and have made thorough revisions based on your guidance. We believe these enhancements will significantly improve the quality and depth of our research, enabling it to make a substantial contribution to the literature.

Thank you again for your professional guidance and valuable time. We look forward to your further feedback and hope our efforts meet your expectations, ultimately allowing our research to make a significant contribution to the relevant fields.

Sincerely,

Li Zhang，Caixia Guo

---

## [Editor Report · Decision Letter 1]

8 Feb 2024

Can corporate ESG performance improve audit efficiency?: Empirical evidence based on audit latency perspective

PONE-D-23-42504R1

Dear Dr. Zhang,

We’re pleased to inform you that your manuscript has been judged scientifically suitable for publication and will be formally accepted for publication once it meets all outstanding technical requirements.

Kind regards,

Simon Grima, PhD

Academic Editor

PLOS ONE

Additional Editor Comments (optional):

suggestions have been taken on board and addressed adequately
---

## [Editor Report · Acceptance letter]

1 Mar 2024

PONE-D-23-42504R1 

PLOS ONE

Dear Dr. Zhang, 

I'm pleased to inform you that your manuscript has been deemed suitable for publication in PLOS ONE. Congratulations! Your manuscript is now being handed over to our production team.

Kind regards, 

on behalf of

Professor Simon Grima 

Academic Editor

PLOS ONE